# Psychophysiological Responses of Cut Flower Fragrances as an Olfactory Stimulation by Measurement of Electroencephalogram in Adults

**DOI:** 10.3390/ijerph191811639

**Published:** 2022-09-15

**Authors:** Yu-Tong Wu, A-Young Lee, Na-Yoon Choi, Sin-Ae Park

**Affiliations:** 1Department of Bio and Healing Convergence, Graduate School, Konkuk University, Seoul 05029, Korea; 2Plant, Environment, and Health Lab, Konkuk University, Seoul 05029, Korea; 3Department of Horticulture and Landscape Architecture, National Taiwan University, Taipei 106032, Taiwan; 4Department of Systems Biotechnology, Konkuk University, Seoul 05029, Korea

**Keywords:** attention deficit, hyperactivity disorder, clinical skill, horticultural therapy, nature experience, orbitofrontal cortex

## Abstract

Horticultural therapy (HT) is green care that can help improve and recover the health of modern people living in cities through natural experiences. Many studies have been conducted to determine HT’s therapeutic effects and underlying mechanisms, but investigation for developing readily applicable clinical techniques is insufficient. We aimed to investigate adults’ brain activity and emotional state during flower arrangement (FA) with different flowers in an HT program. We recruited thirty adults and used a randomized cross-over study method to set them to participate in five FA tasks at 90-s intervals. While performing FA tasks, the participants’ prefrontal cortex brain waves were measured by a wireless electroencephalography device and their emotional states between FA tasks were measured by questionnaires. Results showed that each FA task resulted in a different attention level of the participants. The participants showed the highest attention level during FA with stocks and carnations, while FA with lilies showed the lowest attention level among the five FA tasks. Instead, the participants showed the highest arousal, tension, and anxiety for emotional states during FA with lilies. Therefore, this study confirmed the differences in attention level and emotional changes according to flower types for using clinical techniques of HT for various clients.

## 1. Introduction

Around the potential of the natural environment, more and more attention has been paid to the positive impact that the natural environment can provide on human health and well-being [1]. This trend is supported by theories like the attention restoration theory and psycho-evolutionary stress reduction theory, which claim the natural environment serves as a practical component for mental stability and attention recovery in humans [2,3]. Since more than half of the world’s population lives in urban areas, and modern city life brings about more attention to fatigue and daily stress, the natural environment and its integration into city landscapes and routines have become a remedy for stressed urban populations [1]. Making the situation even worse, the recent global COVID-19 pandemic crisis and resultant home quarantine and isolation, social distancing, and remote working at home cause social and psychological problems, such as anxiety, depression, or other temporary and chronic mental diseases than any other time in the modern history [4].

Under these circumstances, green care has drawn the attention of researchers and practitioners worldwide as an alternative, active process to promote the health and well-being of urban residents. Green care refers to activities designed to provide green and natural environments to humans. This new global trend has been established in recent years in leading agricultural countries in Europe, North America, and Asia [5,6,7]. As a type of green care, horticultural therapy is defined as the use of horticultural activities for clients with special needs in specific therapeutic sessions moderated by a professional therapist [8,9]. Six positive health benefits can be found in horticultural therapy: Physical, psychological, social, cognitive, behavioral, and educational [10]. Horticultural activity can enable modern people to readily interact with natural elements even within a limited urban environment, not restricted by the surrounding physical environment [11]. Examples of horticultural activities used in a therapeutic session include, but are not limited to, gardening outdoors, planting indoors, and making crafts with living plants and flower arrangements [10].

One of the most popular therapeutic horticultural activities is flower arrangement (FA) in which clients are given flowery materials and tools to create or install decorations in the living space. As an extension of nature, FA is a form of art and craft as the beauty of nature itself and the formative skills of the practitioner converge to create an aesthetic object with flowering plants and other natural objects [12]. Moreover, the ingredients in the rose extract can significantly reduce the cognitive impairment of dementia, which was found by Esfandiary et al. [13]. Many types of flowers have distinct scents and visual properties, which can result in health effects to a different extent through the sensory system when used in horticultural therapy programs [14,15]. Therapeutic effects of FA in horticultural therapy programs have been investigated in various aspects: Physical effects (upper limb function, grip strength), cognitive effects (concentration, perseverance, accuracy of task), and psychological effects (psychotic symptoms, self-efficacy, stress) [16,17,18,19].

The therapeutic mechanisms of FA have been revealed through studies using devices that measured physiological responses, such as electroencephalogram (EEG), electromyography (EMG), heart rate variability (HRV), blood pressure, and near-infrared spectroscopy (NIRS) [20,21,22,23,24]. FA using natural flowers was found to increase sympathetic nerve stability and relaxation while reducing negative emotions like depression and anxiety in adult clients, compared with those when artificial flowers were given [21]. Tao et al. [23] reported that a group of women in their 20s who participated in FA showed relaxation effects represented by increased brain alpha waves and reduced blood pressure. In another study, FA also showed effects on cognitive function activation, increased concentration, and relaxation in older adults and older adults with dementia [24,25].

Despite the previous mechanistic findings, there is a lack of information about clinical techniques or the selection of floral materials to maximize the therapeutic effects of FA in coordinating a horticultural therapy program. As a result, most horticultural therapy programs are developed and executed based on the therapist’s speculations or personally conceived ideas, not based on scientific findings [26]. Therefore, the objective of this study was set to investigate the psychophysiological responses of individuals participating in FA using different flowers.

## 2. Materials and Methods

### 2.1. Participants

The participants of this study were a total of thirty adults over 20 years old of age (average age: 34.87 ± 13.04 years). We posted a recruitment notice on bulletin boards at apartments located in Gwangjin-gu, Seoul, South Korea. Moreover, the notice was posted on online bulletin boards of the official website of the government district. The selection criteria of the participants of this study were (1) adults over 20 years old without mental illness, (2) normal olfactory function, and (3) right-handed persons. The third criterion was set to eliminate possible interruptions of the dominance of handedness to the brainwaves. Studies have shown that the human cerebral cortex and cerebellar cortex produce different brainwaves depending on the movement of the dominant and non-dominant hands [27,28]. As caffeine or similar kinds of oral stimulants can disturb brain activity, the participants were required to fast 2 h before the experiment began. In addition, they were asked to complete an olfactory function test, scent survey, and visual analog scale for screening purposes. We also collected demographic information, age, gender, and body composition of the participants.

At the end of the experiment, subjects received $8 as an incentive to complete the experiment. The Institutional Review Board of Konkuk University approved this study (7001355-202103-HR-429).

### 2.2. Experimental Environment

We conducted the experiment in a special indoor space of 220 × 190 cm, on the Seoul campus of Konkuk University, Republic of Korea. The average air temperature, relative humidity, and brightness during the experiment were 24.47 ± 3.62 °C, 37.86 ± 14.01%, and 8285.76 ± 6703.80 LUX. Ivory curtains were arranged in the front and both sides of the participants’ seating positions in the experiment space, while the table was covered with a white tablecloth to help them to focus on the visual stimuli. There were five flowering plants in the experimental design, chrysanthemum (*Dendranthema grandiflora*), lily (*Lilium orientalis Casa Blanca*), hoary stock (*Matthiola incana*), rose (*Hybrid tea rose*), and carnation (*Dianthus caryophyllus Adorable series*).

In consideration of color, we chose to unify the color of the flowers to white in the experiment. And we trimmed flowers and readjusted the flowers’ height to the level of the nose of each participant. We placed the trimmed flowers in a black barrel container (19 × 23 cm) on the left and the flower mud (11 × 7.5 × 14 cm) in an ivory white pot (13 × 15 × 15 cm) on the right (Figure 1).

### 2.3. Experimental Protocol

The participants conducted each FA task for 90 s on average. Once they entered the experiment space, they wore the EEG device and had a rest on the chair before performing FA tasks. Then, they were assigned to perform a FA task for 90 s in random order. After the FA task, the participants were asked to answer a subjective questionnaire to record the changes in their emotions. In sum, this measurement routine consisted of a 30 s rest, a 90 s task, and a survey, and was repeated until the participants completed all five FA tasks we designed (Figure 2) [29]. Experiment participants were told how to perform FA activities before the experiment started. All experimental procedure was conducted in a blinded manner using a randomized cross-over study method, so the participants were requested not to speak or take any unnecessary activity during the experiment.

### 2.4. Measurements

#### 2.4.1. Electroencephalogram (EEG)

A wireless dry EEG device (Quick-8; Cognionics, San Diego, CA, USA) was used to measure the brainwaves of the participants during FA tasks. The dry type of EEG device ensures the safety of the user from a potential electric shock by employing electrolyte gel on the device, which is different from the wet type. The dry electrodes integrated into the scalp of the EEG device amplify and further process measured electrical signals. The Defense Advanced Research Planning Agency has approved the use of this device for this study. The international standard of 10- to 20-electrode arrangement system was used to collect the brainwave data, and the reference electrode was attached to the left earlobe (A1) [30]. The orbitofrontal cortex in the prefrontal cortex (PFC) is the region that converges and integrates olfactory, visual, and gustatory senses [31]. As such, we measured and analyzed EEG results from the left (Fp1) and the right (Fp2) PFC channels (Figure 3).

#### 2.4.2. Subjective Evaluation of the Emotional States

Along with the EEG brainwave data collection, we also used subjective evaluations to verify the EEG results. These were the profile of mood state (POMS) and semantic differential methods (SDM) that are widely accepted in assessing emotional states of individuals.

The POMS questionnaire, originally developed by McNair et al. [32] and translated into Korean by Kim et al. [33], consists of 30 items to measure six emotional state variables: Tension and Anxiety (T-A), Depression (D), Anger and Hostility (A-H), Vigor (V), Confusion (C), and Fatigue (F) [32,34]. These variables are evaluated on a 5-point scale in each question from 1 (Not at all) to 5 (Very much). Thus, lower scores indicate better mental state of respondents. The total mood disturbance (TMD) score was further evaluated using the same response, following the formula: TMD score = (T-A) + (D) + (A-H) + (F) + (C) − (V). Lower TMD scores indicate more positive moods of respondents.

Created by Osgood (1952), SDM is also a method to evaluate emotional states through adjectives. The SDM questionnaire is composed of three items: “Comfortable-uncomfortable”, “natural-artificial”, and “relaxed-awakening”, which are measured on a 13-point Likert scale from −6 to +6. Unlike the POMS, higher scores of SDM present positive emotions of respondents [35].

#### 2.4.3. Scent Strength of Flowers

We used the Labeled Magnitude Scale (LMS) [36] to measure subjective scent strength of flowers. The perceived scent strength by the participants was plotted on 0-to-100-point LMS and categorized into six stages by intensity: No sensation (0–1.4 score), weak (1.4–6 score), moderate (6–17 score), strong (17–35 score), very strong (35–53 score), and strongest imaginable (54–100 score).

### 2.5. Data Analysis

The collected EEG data were analyzed by the Bioteck Analysis Program (Bioteck, Daejeon, South Korea). Across the entire frequency band (4–50 Hz), brain waves from PFC were classified into four regions: Theta (4–8 Hz), alpha (8–13 Hz), beta (13–30 Hz), and gamma waves (30–50 Hz) before processing [37]. These first-hand data can be further parameterized into three indices: Relative fast alpha power spectrum (RFA), relative slow alpha power spectrum (RSA), and sensory-motor rhythm (SMR)/theta spectrum. These second-hand data present different states of the brain in attention (Table 1).

On SPSS (IBM SPSS Statistics 25; Armonk, N.Y., USA), one-way ANOVA was run to examine the effects of FA tasks on the EEG parameters, POMS and SDM scores, and scent strength of flowers. Duncan’s post hoc analysis was used to separate the differences in the means. Statistical significance was set at *p* < 0.05 in all of the comparisons made in the analyses. For the descriptive statistics to report the demographic data of the participants, WPS Office Excel (WPS Office for Windows, V.11.2.0.10101; Kingsoft Powerword, 2021) was used.

## 3. Results

### 3.1. Demographic Information

A total of thirty adults over the age of 20 participated in the study. The purpose of this study was to investigate the effects of olfactory stimuli, and the olfactory function of the experimental participants was tested. The results are shown in Table 2. That means the participants in this experiment have normal olfactory function.

### 3.2. Electroencephalogram (EEG)

The RFA of the left region of PFC was significantly different among five FA tasks (Table 3). During the FA tasks with stock and carnation, the RFA was higher than those during the other FA tasks. In contrast, the RFA during the FA tasks with lily was lower than those during the other FA tasks.

### 3.3. Subjective Evaluation of the Emotional States

The results of the SDM showed that the participants felt natural and relaxed when performing FA tasks, but the emotional states were significantly different among FA tasks (Figure 4). The type of flower plants also affected the subjective emotional states. When performing FA tasks with chrysanthemum, rose, and carnations, the participants experienced the most natural and relaxed feelings over the other FA tasks. On the other hand, when performing FA tasks with lily, the participants experienced less natural and awakening feelings (Figure 4).

The results of the POMS showed that only T-A scores were significantly different between FA tasks. The participants showed the greatest T-A scores during FA tasks with lily, while no differences were observed in the TMD scores, V, F, C, A-H, and D indices (Figure 5 and Figure 6).

### 3.4. Scent Strength

Roses, chrysanthemums, carnations, and stocks resulted in a strong scent perceived by the participants in our LMS classification, all measured around 20. These values were significantly different from the score measured in lilies, which recorded the strongest imaginable over 50 (Figure 7).

## 4. Discussion

We characterized flower arrangements (FA) using five different types of flowers with the measurements of brain activity and emotional states of the participants. The key EEG results showed that the RFA of the left region of PFC, which is related to concentration with calmness and relaxation of the participants, was highly activated during the FA tasks with stocks and carnations, compared with the other flowers. This difference in brain activity in the PFC is thought to be derived from flowers’ scent and shape. Each flower has distinguished olfactory and visual stimuli conveyed throughout our sensory system, processed uniformly, and are eventually integrated into the processes of the PFC. The processed information leads to differences in physiological responses, which are reflected by the signature brain wave patterns [31,43,44].

The FA participants using carnations and stocks showed the highest concentration in the stable state. Carnations and stocks are popular flowers and the demand for these flowers comes from their interesting flower shapes and attractive scents [45,46]. Moreover, these two flowers have a common volatile substance and scent similar to cloves named ‘eugenol’ [45,46,47]. Cloves are the most highly identified floral aromas worldwide and are widely used for various purposes ranging from perfume to medicine. Psychological studies reflected when subjects worked in a fragrant indoor environment, self-confidence and work efficiency were both escalated [48]. Furthermore, the other study also reflected that flower fragrance can induce brain activity, which affects human activity, language, and memory function [43]. In addition, the scent of cloves has been reported to be effective in reducing restlessness and headaches [49].

Lilies are also one of the most popular cut flowers worldwide. But in this study, when the participants used lilies to perform the task, the experimental participants showed the lowest rest concentration among the five flowers, which means that the fragrance of lilies made the participants more relatively awakened. The olfactory stimuli induce physiological responses, like relaxation and arousal, depending on the intensity [50] (Figure 7). However, this strongest imaginable scent of lilies also can cause an adverse effect [51]. The main aromatic ingredients of lilies are 3,7-dimethyl linalool, homomenthyl salicylate, and others [52]. As these volatile compounds are released at a high level, producing a solid scent to the surroundings, they may cause arousal effects, making the subjects less stable than the other FA tasks [53]. The volatile substance of lily, ‘linalool,’ of lily aroma oil, had physiological arousal and stimulating effects, such as increasing β waves in the brain, galvanic skin response (GSR), and sympathetic nervous activity of the participants [52]. Other studies have shown that the α wave in the left brain decreases in response to lily aroma [54].

A significant difference was found between FA tasks in POMS and SDM scores. Especially, the participants responded that they felt relatively less natural and stable, while conducting FA tasks using lilies, indicating that tension and anxiety were higher than in FA tasks using the other flowers. Because the sensory system influences the induction of emotions [43,44], flowers’ olfactory and visual characteristics provide discerned stimuli to the brain, resulting in the psychophysiological response. Indeed, odors influence mood, evoke powerful sensory experiences of pleasure or displeasure and trigger long-forgotten emotional memories [55]. Similar to our results, Hui-Tang et al. [52] showed that most people chose commendatory terms (e.g., excited and enjoyable) to describe the lily aroma.

The differences in brain activation state and emotional changes during these five FA tasks could likely help horticultural therapy professionals to develop and apply more effective horticultural therapy programs by strategically targeting the subjects. For instance, it would be most effective to use carnations and stocks in therapeutic FA programs for clients like children, adolescents, and those with ADHD or autism because these clients need attention development and improvement [56,57]. In contrast, it would not be more appropriate to apply FA tasks using lilies in a client with a depression and anxiety disorder or under high-stress conditions with excessive arousal [24,57].

However, we also identified the limitations of the study. First, we had not considered the baseline of the participant’s brain activity by measuring it before they began FA tasks. Second, we could not control other factors and variables that might have affected the results, such as tactile, visual, and motor stimuli during FA tasks. Finally, the present study found that the positive psychophysiological effects of FA were derived from the olfactory stimulation through reviewing and discussing previous studies’ results. Thus, to improve the study and increase the reliability of the study, more well-controlled study designs should be employed to conduct scientifically solid research.

## 5. Conclusions

This study showed that the five FA tasks with different types of flowers stimulated the brain activity in the prefrontal cortex differently and also changed the emotional status of the participants performing the FA tasks. Carnations and stocks were the most effective in the participants’ brain activation for concentration in the stable state. On the other hand, FA tasks with lilies caused brain activation in arousal states and emotions of tension and anxiety. The results of this study could be used as reference data for the development and coordination of professional therapeutic programs of horticultural therapy for various clients. As the sensitivity of the human sensory system is also affected by gender, investigation related to gender responses should follow the present study. Furthermore, data for a broader range of client groups are required to ensure HT programs’ evidence-based development.

## Figures and Tables

**Figure 1 ijerph-19-11639-f001:**
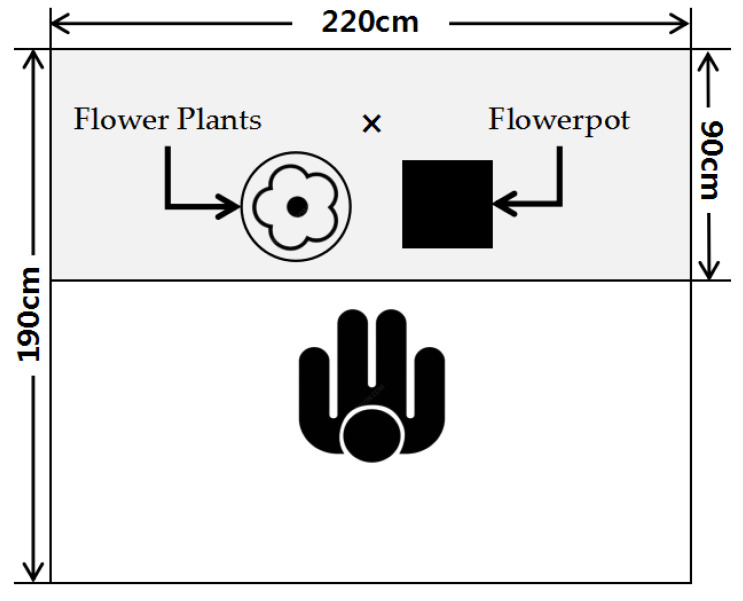
Experimental space layout.

**Figure 2 ijerph-19-11639-f002:**
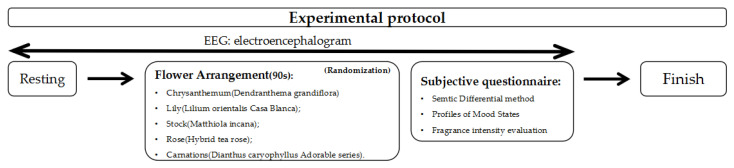
Experiment protocol.

**Figure 3 ijerph-19-11639-f003:**
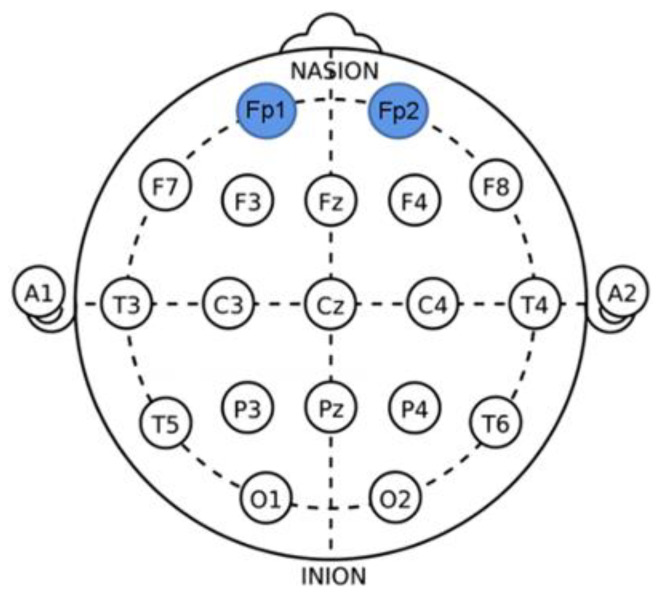
Electrode locations of the electroencephalography (EEG) for this experiment.

**Figure 4 ijerph-19-11639-f004:**
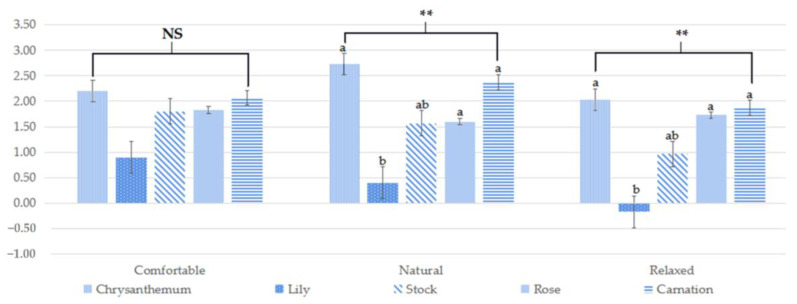
The results of semantic differential method (SDM) evaluation when performing five activities (** is *p* < 0.01, a > b, NS = no significant difference).

**Figure 5 ijerph-19-11639-f005:**
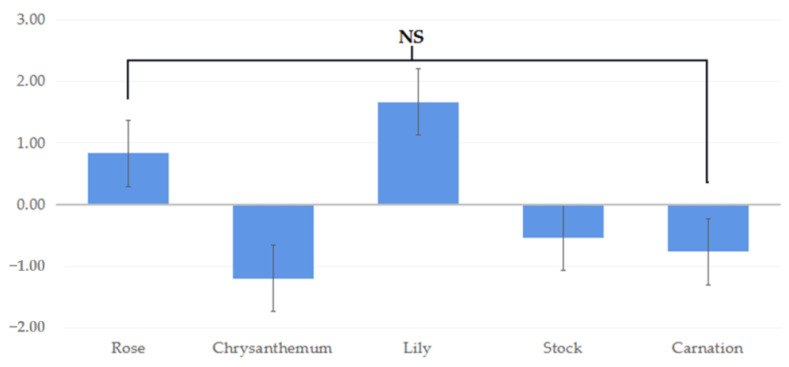
The result of total mood disturbance (TMD) is the total score on the profile of mood state (POMS) when performing five activities (NS = no significant difference).

**Figure 6 ijerph-19-11639-f006:**
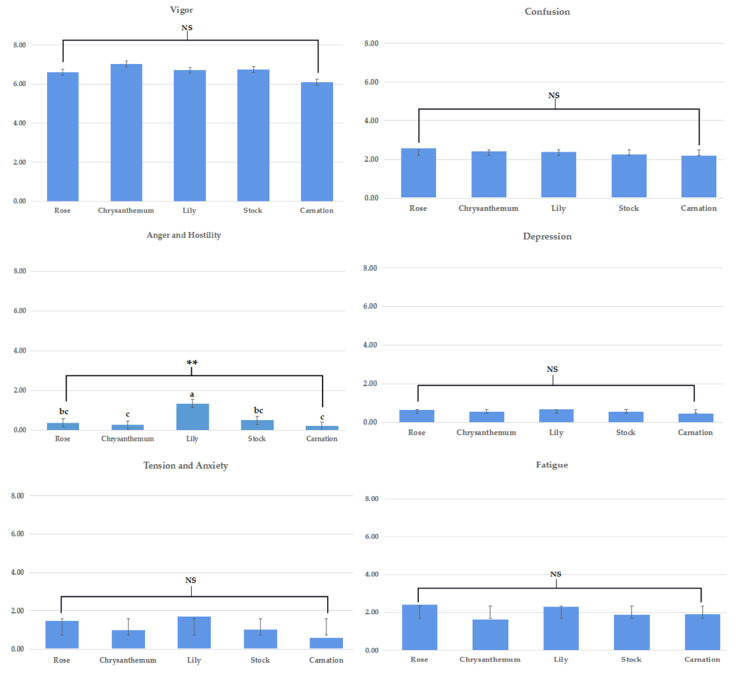
The results of profile of mood state (POMS) are divided into six indicators: Vigor, confusion, anger and hostility, depression, tension and anxiety, fatigue (NS = no significant difference, ** is *p* < 0.01, a > b > c).

**Figure 7 ijerph-19-11639-f007:**
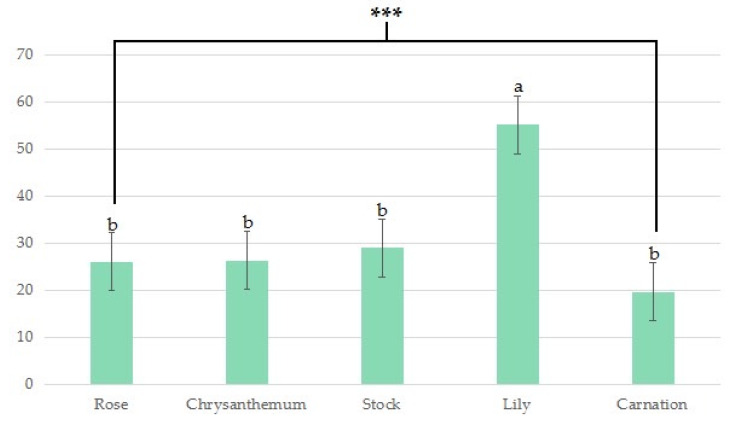
Scent strength of five different types of flowers; no sensation (0 score), weak (1–5 score), moderate (6–15 score), strong (16–35 score), very strong (36–50 score), and strongest imaginable (51–100 score) (*** is *p* <0.001, a > b).

**Table 1 ijerph-19-11639-t001:** EEG power spectrum indicators used in this study.

Analysis Indicator	The Full Name of the EEG Power Spectrum Indicator	Indicator Estimate (Ratio)	State
**RFA**	Relative fast alpha	Higher alpha (11–13 Hz)/Total frequency (4–50 Hz)	Attention and concentration in a relaxed state [38,39].
**RSA**	Relative slow alpha	Lower alpha (11–13 Hz)/Total frequency (4–50 Hz)	Stability and relax [38,39].
**SMR/theta**	Sensory Motor Rhythm to theta	Lower beta (12–15 Hz)/theta (4–8 Hz)	Attention related to cognitive function [40].

**Table 2 ijerph-19-11639-t002:** Descriptive characteristics of the participants (N = 30).

Variable		Male (n = 7)	Female (n = 23)	Total (N = 30)
		Mean ± SD		
**Age (years)**		32.57 ± 12.46	35.57 ± 13.41	34.87 ± 13.04
**Body Height (cm)**		175.94 ± 3.36	162.16 ± 4.33	165.61 ± 7.30
**Body Weight (kg)**		70.94 ± 13.86	57.56 ± 9.80	60.91 ± 12.19
**Olfactory** **Function**	**SSS ^1^**	85.71 ± 7.89	83.7 ± 13.66	83.77 ± 12.47
	**VAS ^2^**	8.57 ± 0.98	7.78 ± 1.62	7.97 ± 1.51

^1^ SSS = Scent survey for screening; A score of 47 or higher is a normal olfactory group [41,42]. ^2^ VAS = Visual analog scale; A score of 5 or higher is a normal olfactory group [41,42].

**Table 3 ijerph-19-11639-t003:** Results of EEG values of RFA (Relative Fast Alpha), SMR (Sensorimotor Rhythm), and RSA (Relative Slow Alpha) indicators according to the activity using each plant (N = 30).

Flower Arrangement Activity		RFA ^1^		SMR ^2^		RSA ^3^	
		Fp1 ^6^	Fp2 ^6^	Fp1	Fp2	Fp1	Fp2
		Mean ± SD					
**Total** **(N = 30)**	**Chrysanthemum**	0.048 ± 0.006 ab^5^	0.049 ± 0.008	0.060 ± 0.007	0.064 ± 0.012	0.121 ± 0.016	0.122 ± 0.017
	**Lily**	0.044 ± 0.004 b	0.049 ± 0.009	0.057 ± 0.009	0.062 ± 0.013	0.113 ± 0.022	0.116 ± 0.020
	**Stock**	0.052 ± 0.013 a	0.051 ± 0.007	0.065 ± 0.015	0.066 ± 0.013	0.118 ± 0.021	0.118 ± 0.015
	**Rose**	0.048 ± 0.007 ab	0.049 ± 0.009	0.060 ± 0.009	0.062 ± 0.011	0.118 ± 0.024	0.116 ± 0.016
	**Carnation**	0.050 ± 0.011 a	0.049 ± 0.008	0.063 ± 0.018	0.061 ± 0.010	0.121 ± 0.027	0.119 ± 0.014
**P ^4^**		0.024 *	0.886 ^NS^	0.157 ^NS^	0.518 ^NS^	0.508 ^NS^	0.676 ^NS^

^1^ The frequency of Relative Fast Alpha (RFA) [(11 Hz–13 Hz)/(4 Hz–50 Hz)] is an indicator of concentration. ^2^ The frequency of SMR (Sensorimotor Rhythm) [(12 Hz–15 Hz)/(4 Hz–50 Hz)] is an indicator of concentration. ^3^ The frequency of RSA (Relative Slow Alpha) [(8 Hz–13 Hz)/(4 Hz–50 Hz)] is an indicator of emotional stability. ^4^ Statistical method used one-way ANOVA (^NS^ had no significant difference, * *p* < 0.05). ^5^ The statistical method used Duncan’s post hoc analysis (a > b). ^6^ Fp1: left frontal lobe; Fp2: right frontal lobe.

## Data Availability

The datasets generated for this study are available on request to the corresponding author.

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
