# Peer review of "Psychophysiological Responses of Cut Flower Fragrances as an Olfactory Stimulation by Measurement of Electroencephalogram in Adults"

_ijerph, 2022, doi:10.3390/ijerph191811639_

Round 1
Reviewer 1 Report
I would recommend you to revise the study and consider obtaining baseline assessments for comparison with the post-FA scores. The first sentence of the introduction seems plagiarism from the cited paper and would avoid this for the rest of the paper (the sentence on line 33 and 34 is exactly used from the introduction in the citation 1). While there were significant differences, on individual poms scales, total score was not significant and the differences were modest and may not suggest big difference in clinical practice. Make sure to provide citation for the Table 1 (how are the Quantitative EEG brain waves associated with the attentional tasks).
Author Response
Thank you for your constructive suggestions on different aspects of the manuscript's contents. Revisions to the reviewer's suggestions are indicated in blue in the manuscript.
First, the primary purpose of our research is to compare the differences in scent stimulation given by flowers as primary data for developing therapeutic skills for horticultural therapy. Previous studies suggested that flower arranging and horticultural activities resulted in a difference in brain wave activity, indicating improvement in concentration. Therefore, baseline measurement had not been carried out to achieve the desired results. However, we are well aware of the limitations of the research design of this study, and the limitations of the study are additionally specified in the conclusion of this study.
Second, the introduction's first sentence has been modified as follows: Around the potential of the natural environment, more and more attention has been paid to the positive impact that the natural environment can provide on human health and well-being.
Third, On the subjective questionnaire poms, as stated in the research method, Poms have six emotional state variables: Tension and Anxiety (T-A), Depression (D), Anger and Hostility (A-H), Vigor (V), Confusion (C ), and Fatigue (F), and the total mood disturbance (TMD). So although there is no significant difference in the TMD, there is still a significant difference in Anger and Hostility (A-H). Combined with the Labeled Magnitude Scale, it can be concluded that the solid olfactory stimulation of lily evoked anger in the experimental participants.
Lastly, we added Table 1 for references related to EEG and attention. Thanks for the excellent point.

Reviewer 2 Report
There are problems with research design.
The flower arrangement of the experiment is unknown.
Although it is an olfactory stimulus experiment, flower arranging is a work activity, and stimuli other than olfactory stimuli are mixed. Visual, tactile, or exercise effects.
Furthermore, brain activity in flower arrangements differs greatly between subjects who are interested in flowers and those who are not.
It is not possible to evaluate only the olfactory stimulus in the unknown flower arrangement, so it cannot be published.
Author Response
There are problems with research design.
The flower arrangement of the experiment is unknown.
Although it is an olfactory stimulus experiment, flower arranging is a work activity, and stimuli other than olfactory stimuli are mixed. Visual, tactile, or exercise effects.
Furthermore, brain activity in flower arrangements differs greatly between subjects who are interested in flowers and those who are not.
It is not possible to evaluate only the olfactory stimulus in the unknown flower arrangement, so it cannot be published.
First of all, we are very grateful for the critical review that the reviewer has made on our study. In response to the suggestions and comments, we have revisions in green in the manuscript.
We fully agree with the reviewers' comments. We acknowledge that the study has limitations in that we did not control all the various stimuli that could interfere with how the participants would feel during flower arranging activities. We mentioned this as an additional study limitation in the review section. When using the five flowers, we tried to unify the color of the flower, the length of the branch, and the arrangement of the flowers. Nevertheless, we admit that it has little control over other variables. We believed that the difference in the characteristics of EEG activity among the FA using these five flowers was due to the olfactory stimulation through reviewing the results of previous studies. More reliable results with comparable control groups will be obtained in our future research.
In the reviewer's comments, it is also mentioned that in this experiment, the influence of olfactory stimulus and visual, tactile, and exercise effects might come into play. Therefore, we also considered the influence of relevant factors in the experimental design stage, so we made certain considerations in the selection of materials required for the activity as follows.
First, for the visual aspect, we chose to unify the color to white, which is also described in this article (please refer to the content on page 3, line 121).
Second, in terms of the sense of touch, and tactile effects, all experimental materials that the participants would touch during the experiment were unified. Concerning flower selection, the differences between the five kinds of flowers were imperceptible.
For the exercise effects, we had the same opinion as the reviewer. Although the FA are not high-intensity agricultural activities, the experimental participants still can be affected by the active engagement of the muscles during the FA. Nonetheless, this study is an experimental study to explore the effects of five different Olfactory Stimulation on adults through the FA. During each kind of flower for FA, it was carried out in the same position and used simultaneously, so we believe there was no difference in the movement effects depending on the five kinds of flowers.
To sum up, we do not disagree that FA activity has a certain influence on visual, tactile, and motor effects, but there is no difference in the 5 FA activities. We believe that all experimental participants felt the same stimuli in the above three elements.
Regarding the differences caused by the preference for flowers the reviewer mentioned, there is no relevant investigation or research on the preference in this experiment. However, perhaps this can be the direction of the next experimental research.
In addition, although it is only our subjective opinion, the participants in this experiment are mainly recruited through the Internet. We suppose Therefore, if they have no interest in the flower, maybe they will not choose to become an experimental participant.
We hope our answers can resolve the concerns the reviewer raised. Please let us know if there are any deficiencies in our paper.
Again, we thank you for the review.

Reviewer 3 Report
Thank you for the opportunity to review your manuscript, which reports an experimental study exploring differential effects of flower arrangement across 5 different flower types. The study appears to have been conducted according to an appropriate design to address the aim, which was to ascertain differences in attentional levels and emotional state changes according to the different flower types. As a feature of horticultural therapy, knowledge about differential effects from the flowers used for therapeutic purposes has important clinical implications and applications.
I have inserted highlights within the manuscript either to indicate where attention needs to be directed to wording, or to flag some specific comments or queries for you to take into consideration.

Author Response
We appreciate the reviewer's positive review of this experimental study; we have revised the manuscript upon the reviewer's suggestions, marked in red in the text.
Regarding the question about the length of the experiment, the following is our response.
First of all, when we conceived and designed the experiment, we mainly referred to papers related to olfactory stimulation studies. Usually, in experiments in the field of aromatherapy, the duration of olfactory stimulation is set to 90 sec.
(Ikei, H et al. Comparison of the effects of olfactory stimulation by air-dried and high-temperature-dried wood chips of hinoki cypress (Chamaecyparis obtusa) on prefrontal cortex activity. Journal of Wood Science, 2015, 61, 537-540; DOI: 10.1007/s10086-015-1495-6
Igarashi, M et al. Effects of olfactory stimulation with rose and orange oil on prefrontal cortex activity. Complementary therapies in medicine, 2014, 22, 1027-1031, DOI: 10.1016/j.ctim.2014.09.003)
Secondly, there are five kinds of flowers used in this experiment, and the test of the supervisor questionnaire was carried out after each FA activity, which prolonged the time of the experiment. Excessive experimental time would not only affect the enthusiasm of experimental participants but also have a certain impact on the accuracy of EEG. Moreover, we also conducted simulation experiments before the formal experiments started to ensure that the formal experiments would not result in the situation that the FA activities could not be completed or ended earlier.
Lastly, any grammar or minor corrections the reviewer suggested in the manuscript have been corrected. Thanks for again for the thoughtful review.
We hope the above answers your inquiries about the study.

Round 2
Reviewer 1 Report
Minor correction: Please describe the type of study in abstract for eg case control vs cohort study, cross-sectional etc to give the summary to readers.
However, having just put the study method deficiencies such as not having a baseline and not controlling for other stimuli besides olfactory as a weakness, does not add to novelty and current knowledge of the literature.
Author Response
Thank you very much for your thoughtful review. We have revised the manuscript upon your suggestions about the type of research in the abstract, which has been marked in blue for your review.
As for the deficiencies in the research methods you mentioned, we admit that this study was not designed as a complete experiment, leaving limitations to the interpretation of the results. We plan to amend the design and confirm what we found in the study in our future research.
Again, we genuinely appreciate your attention and suggestions on this article.
